# Towards Efficient Post-training Quantization of Pre-trained Language Models

**Haoli Bai**[1,2*]**, Lu Hou**[1]**, Lifeng Shang**[1]**, Xin Jiang**[1]**, Irwin King**[2]**, Michael R. Lyu**[2]
[1]Huawei Noah's Ark Lab,    [2]The Chinese University of Hong Kong
{baihaoli,houlu3,Shang.Lifeng,Jiang.Xin}@huawei.com,
{king,lyu}@cse.cuhk.edu.hk

## Abstract

Network quantization has gained increasing attention with the rapid growth of large pre-trained language models (PLMs). However, most existing quantization methods for PLMs follow quantization-aware training (QAT) that requires end-to-end training with full access to the entire dataset. Therefore, they suffer from slow training, large memory overhead, and data accessibility issues. In this paper, we study post-training quantization (PTQ) of PLMs, and propose module-wise quantization error minimization (MREM), an efficient solution to mitigate these issues. By partitioning the PLM into multiple modules, we minimize the reconstruction error incurred by quantization for each module. In addition, we design a new model parallel training strategy such that each module can be trained locally on separate computing devices without waiting for preceding modules, which brings nearly the theoretical training speed-up (e.g., $4\times$ on $4$ GPUs). Experiments on GLUE and SQuAD benchmarks show that our proposed PTQ solution not only performs close to QAT, but also enjoys significant reductions in training time, memory overhead, and data consumption.

## 1  Introduction

Large pre-trained language models (PLMs) have achieved remarkable success in various natural language processing tasks [44, 12, 5]. However, the increasing size and computation overhead also make it prohibitive to deploy these PLMs on resource-constrained devices. To obtain compact PLMs, various model compression methods have been proposed, such as pruning [32, 14, 50], knowledge distillation [40, 42, 24], weight-sharing [10, 25, 46, 22], dynamic computation with adaptive depth or width [18, 52, 59], and quantization [55, 41, 54, 56, 3, 38, 43].

Among these methods, network quantization enjoys the reduction of both model size and computation overhead without modifying the network architecture, and is thus extensively studied in PLM quantization [55, 41, 54, 56, 3, 38, 43]. However, most previous methods for PLM quantization methods follow quantization-aware training (QAT), which suffers from several issues. Specifically, QAT usually conducts end-to-end back-propagation training over the whole training set, which can be slow in training time, memory demanding and data consuming. These issues can be sometimes prohibited for industrial PLMs. Post-training quantization (PTQ), on the other hand, serves as an appealing alternative that is efficient in training time, memory overhead and data consumption. Instead of the full training set, PTQ can leverage only a small portion of training data to minimize the layer-wise reconstruction error incurred by quantization [34, 35, 33, 23]. This can be done by either calibrating the batch normalization statistics [34] or step sizes [35] in quantization functions. The layer-wise objective also breaks down the end-to-end training, making the prob-

---

*This work is partially done during internship at Huawei Noah's Ark Lab.

lem more sample-efficient [58] and memory-saving. Nonetheless, we find it non-trivial to directly apply previous PTQ methods for PLMs such as BERT [12], as the performance drops sharply.

In this paper, we aim at improving the performance of post-training quantization for PLM, while simultaneously maintaining its efficiency w.r.t training time, memory overhead and data accessibility. First, we partition the PLM into multiple modules, and propose *module-wise reconstruction error minimization* (MREM). Each module consists of multiple Transformer layers, which permits the sufficient optimization than the previous layer-wise objective. Meanwhile, the module size can be flexibly adjusted for the proper trade-off between layer-wise correlation and memory overhead of computing devices. While a similar block-wise objective is previously considered in [29], they require second-order Hessian matrices for optimization, which can be computationally prohibitive for large PLMs. Second, we design a new *model parallel strategy* to further accelerate the training process of MREM. By allocating each partitioned module to the individual computing device, our design allows all modules to be trained jointly without synchronizing with adjacent partitions. This brings nearly the theoretical speed-up (e.g., $4\times$ on 4 GPUs), superior to previous data paral-

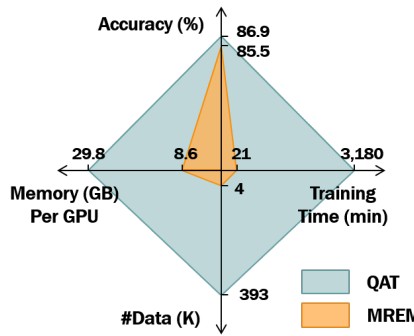

Figure 1: An illustrative comparison between our parallel post-training quantization method (MREM) and QAT on four dimensions of the quantization pipeline: accuracy, training time, memory overhead, and data consumption. The results are based on a quantized BERT-large model with 4-bit weights and 8-bit activations over the MNLI dataset. Best viewed in color.

lel [9, 27] or model parallel [21, 36] techniques. Third, we develop *annealed teacher forcing* for our parallel strategy. We find that the naive parallel training suffers from the propagation of reconstruction error, since each quantized module passes the error to its successors before it is fully optimized. Inspired by [51], we use the full-precision module to provide clean signals to the next quantized module. This breaks the reconstruction error propagation and further improves the performance of parallel strategy.

We summarize the contributions of this paper as follows (i) We study the post-training quantization of PLMs, and propose module-wise reconstruction error minimization (MREM), a fast, memory-saving, and data-efficient approach to improve the quantized PLMs. (ii) We design a new model parallel strategy based on MREM to accelerate post-training quantization with theoretical speed-up for distributed training. (iii) The parallel MREM can be combined with annealed teacher forcing to alleviate the propagation of reconstruction error and boost the performance. iv) Empirical results on the GLUE and SQuAD benchmarks demonstrate superiority of our PTQ solution over QAT. For instance, Figure 1 shows that the BERT-large model trained by parallel MREM achieves $85.5\%$ accuracy with only 4K training samples. Moreover, it consumes merely one-third of memory per GPU and is more than $150\times$ faster than previous QAT training.

## 2 Motivation

### 2.1 Quantization Background

Network quantization replaces the original full-precision weight or activation $\mathbf{x} \in \mathbb{R}^{m \times n}$ with its lower-bit counterpart $\hat{\mathbf{x}}$. Denoting $s \in \mathbb{R}^+$ as the step size, the $b$-bit symmetric uniform quantization function $\mathcal{Q}_b(\cdot)$ can be written as

$$\hat{\mathbf{x}} = \mathcal{Q}_b(\mathbf{x}) = s \cdot \Pi_{\Omega(b)}(\mathbf{x}/s), \tag{1}$$

where $\Omega(b) = \{-2^{b-1} + 1, ..., 0, ..., 2^{b-1} - 1\}$ is the set of $b$-bit integers, and $\Pi(\cdot)$ is the projection function that maps $\mathbf{x}/s$ to its closest integer.

To quantize Transformer-based PLMs, we follow the setting in previous works [55, 56, 3]: we quantize both the network weights and activations in each matrix multiplication. We use symmetric uniform quantization for weights, embeddings, and activations, except activations after the self-attention and GeLU function. For these two activations, we adopt asymmetric quantization since their elements are mostly positive. We skip the quantization for all layer-normalization layers, skip connections, biases

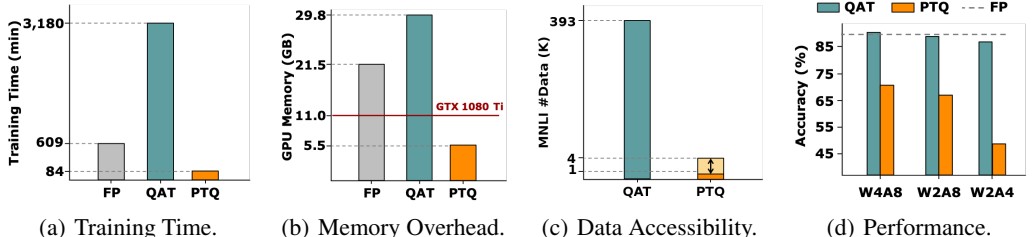

(a) Training Time.  (b) Memory Overhead.  (c) Data Accessibility.  (d) Performance.

Figure 2: Comparison between QAT and REM-based PTQ over four dimensions. We use a BERT-large model over MNLI dataset for illustration. The full-precision (FP) fine-tuning is also included as a baseline. We follow the procedure in [56] for QAT, and REM in Equation (2) for PTQ. The training time and memory in (a) and (b) are measured by 4-bit weights and 8-bit activations (i.e., W4A8) on an NVIDIA V100 GPU.

and the last classification head due to little computation overhead or large performance drop. We leave more details on the choice of quantization functions and their algorithmic details in Appendix A.

**Post-training Quantization.** Recently popular PTQ methods [35, 47, 33, 23, 30] constructs an tiny unlabeled dataset (a.k.a. calibration set) $\tilde{\mathcal{D}}$ from the original training data $\mathcal{D}$. These approaches usually aim at reconstruction error minimization (REM), i.e., minimizing the distance between the output of multiplication between the quantized and the full-precision counterpart as follows:

$$\min_{\mathbf{w},\mathbf{s}} \|\hat{\mathbf{w}}^{\top}\hat{\mathbf{a}} - \mathbf{w}^{\top}\mathbf{a}\|^2, \quad \text{s.t. } \hat{\mathbf{w}} = \mathcal{Q}_{b_w}(\mathbf{w}), \quad \hat{\mathbf{a}} = \mathcal{Q}_{b_a}(\mathbf{a}), \tag{2}$$

where $\mathbf{w}$ and $\mathbf{a}$ are full-precision weights and activations, $\hat{\mathbf{w}}$ and $\hat{\mathbf{a}}$ are their quantized representations with $b_w$ and $b_a$ bit-widths, and $\mathbf{s}$ denotes all step-sizes involved for quantization. REM is usually conducted in a greedy manner. It proceeds to the matrix multiplication only after the training of previous ones. The simple objective in Equation (2) is lightweight to solve over $\tilde{\mathcal{D}}$, and is also theoretically verified to be more sample-efficient than end-to-end training [58]. While there are also other data-free PTQ variants [57, 34, 16, 54], REM-based solutions usually demonstrate better empirical results [33, 23]. Thus we mainly focus on the REM-based PTQ for PLMs in this paper.

## 2.2 Quantizing Pre-trained Language Models: QAT or PTQ?

To conceptually study the difference between QAT and REM-based PTQ over popular PLMs, we consider the following four dimensions, and summarize the results in Figure 2.

**Training Time.** As QAT conducts training over the full training set $\mathcal{D}$, it takes much more time than PTQ over the calibtration set $\tilde{\mathcal{D}}$. Recent QAT methods [56, 3] further combine two-stage knowledge distillation [24], which can take nearly four times longer than FP, as shown in Figure 2(a).

**Memory Overhead.** The increasing size of large PLMs makes it prohibited to conduct QAT on memory-limited devices. From Figure 2(b), QAT [56] even consumes 8.3 GB more memory than FP when armed with knowledge distillation. On the contrary, PTQ only caches intermediate results in Equation (2), which can be fed into a single GTX 1080 Ti.

**Data Accessibility.** Data accessibility can be often prohibited due to data security or privacy issues in the industry. Unlike QAT that needs the full training set $\mathcal{D}$, PTQ constructs the calibration data $\tilde{\mathcal{D}} \subseteq \mathcal{D}$ by sampling only $1\text{K} \sim 4\text{K}$ instances from $\mathcal{D}$, as shown in Figure 2(c).

**Performance.** QAT usually demonstrates superior quantization results than PTQ, and this holds true for PLMs. From Figure 2(d), QAT remains steady across different bit-widths, while REM-based PTQ drops sharply. Thus the poor performance of PTQ is the main challenge in quantizing PLMs.

In summary, REM-based PTQ is superior to QAT with regard to training efficiency, memory overhead, and data accessibility. Nevertheless, it is still often less preferred than QAT due to its severe performance drop especially for low quantization bit-width [55, 41, 56].

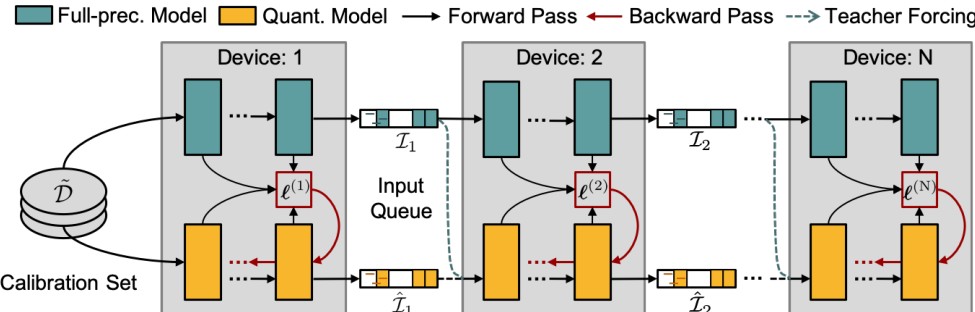

Figure 3: The overview of the proposed module-wise reconstruction error minimization (MREM). We partition both the full-precision model and quantized model into multiple modules and put these modules on different computing devices. By sampling tensors from the input queue, each module can be trained locally without waiting for its predecessors. Teacher forcing is applied to mitigate the issue of reconstruction error propagation on the quantized module.

## 3 Methodology

In this section, we aim at improving the performance of post-training quantization for PLMs, while preserving its merits of fast training, light memory overhead, and data consumption. We extend the existing reconstruction error minimization from the layer-wise to the module-wise granularity to fit Transformer models. Based on the module partition, we then design a new parallel training strategy that further speeds up the PTQ pipeline. An overview of our solution can be found in Figure 3.

### 3.1 Module-wise Reconstruction Error Minimization

We propose a new PTQ solution called *module-wise reconstruction error minimization* (MREM) for PLMs. Existing REM [33] solves Equation (2) for each matrix multiplication. However, a standard transformer layer in PLMs consists of a Multi-Head Attention (MHA) and a Feed-Forward Network (FFN), both of which contain a number of matrix multiplications that are coupled together. Greedily tackling each matrix multiplication in REM can thus lead to suboptimal quantized networks. Moreover, the insufficiently minimized reconstruction error shall propagate and enlarge along with transformer layers, and finally deteriorate the network output [6, 2].

Towards that end, the proposed module-wise reconstruction error minimization admits larger granularity by jointly optimizing all the coupled linear layers inside each module. Specifically, given a PLM with $L$ transformer layers, embedding layers and the classification head, we partition them into $N$ modules, where the $n$-th module include $[l_n, l_{n+1})$ transformer layers with $l_n$ being the first layer of this module[2]. MREM aims at minimizing the joint reconstruction errors between all quantized FFN output $\hat{\boldsymbol{f}}_l$ in the module from their full-precision counterpart $\boldsymbol{f}_l$ as follows:

$$\min_{\mathbf{w}_n, \mathbf{s}_n} \ell^{(n)} = \sum_{l \in [l_n, l_{n+1})} \|\hat{\boldsymbol{f}}_l - \boldsymbol{f}_l\|^2, \tag{3}$$

where $\mathbf{w}_n$ and $\mathbf{s}_n$ are all learnable parameters and quantization step sizes in the $n$-th module. Similar to REM, MREM can be optimized sequentially: given previously trained modules, only parameters and quantization step sizes in the current module are updated. Besides the grouped Transformer layers, we also minimize the MSE loss in the Transformer embedding and output logits respectively.

Note that the number of modules $N$ can be adjusted depending on the memory constraint of computing resources. When $N = 1$, this reduces to intermediate-layer knowledge distillation [24], which can be memory-demanding when quantizing large PLMs on a single GPU.

### 3.2 Accelerated Parallel Training

Based on the proposed MREM, we propose a new model parallel strategy to further accelerate the training. As shown in Figure 3, we put different modules on individual computing devices. A set of

---

[2]The embedding layers and the classification head are incorporated in the first and last module respectively.

input queues $\mathcal{I} = \{\mathcal{I}_1, ..., \mathcal{I}_{N-1}\}$ is deployed between each pair of adjacent modules. For the $n$-th module, the queue collects its output of the most recent $t_0$ steps, i.e., $\mathcal{I}_n^t = \{\boldsymbol{f}_{l_n}^t, \boldsymbol{f}_{l_n}^{t-1}, ..., \boldsymbol{f}_{l_n}^{t-t_0+1}\}$. Meanwhile, the $(n+1)$-th module can always sample with replacement $\boldsymbol{f}_{l_n} \sim \mathcal{I}_n^t$ from the queue without waiting for the $n$-th module. Similar rules hold for the quantized module and their input queues $\hat{\mathcal{I}}$. The design of the input queue resembles *stale synchronous parallel* [17] that caches the stale output locally so as to reduce the waiting time among workers, where $t_0$ is the stale threshold.

The training workflow is as follows. Initially, every module is computed one after another in the first $t_0$ step to fill in the input queue, after which parallel training takes place. Then the module samples input from the queue and calculates the loss $\ell^{(n)}$ correspondingly for $n = 1, ..., N$. Meanwhile, the input queue is updated with the rule of *first-in-first-out* throughout the training. In the backward pass, we constrain the gradients to propagate locally within each module, without affecting its predecessors. Such a design can avoid the load imbalance issue from straggler modules, bringing nearly the theoretical $N\times$ speed-up.

The proposed parallel training strategy is superior to the conventional data parallel training in quantizing PTMs. Meanwhile, it is also different from pipeline parallelism [21]. We leave the detailed comparisons with these parallelism techniques in Appendix C.

### 3.3 Annealed Teaching Forcing

Since all modules get optimized simultaneously instead of the sequential manner, the next module takes the output from the queue before its predecessor is fully optimized. Therefore, the predecessor's reconstruction error is propagated to the following modules before it is sufficiently minimized.

Inspired by teacher forcing [51] in training recurrent networks, the output $\boldsymbol{f}_{l_n}$ from the $n$-th full-precision module naturally serves as the clean input to the $(n+1)$-th quantized module to substitute $\hat{\boldsymbol{f}}_{l_n}$. Thus $\boldsymbol{f}_{l_n}$ stops the propagation of the accumulated error on the quantized module. Nevertheless, such an approach breaks the connection to previous quantized modules and may suffer from forward inconsistency between training and inference [2] on the quantized model. For the trade-off, we take the convex combination between the full-precision $\boldsymbol{f}_{l_n}$ and quantized $\hat{\boldsymbol{f}}_{l_n}$ as follows:

$$\tilde{\boldsymbol{f}}_{l_n} = \lambda \boldsymbol{f}_{l_n} + (1 - \lambda)\hat{\boldsymbol{f}}_{l_n}, \quad \lambda \in [0, 1], \tag{4}$$

where the hyperparameter $\lambda$ controls the strength of teacher forcing. $\lambda = 1$ gives the full correction of reconstruction error but with forward inconsistency, while $\lambda = 0$ reduces to the conventional setting that suffers from the propagated reconstruction error. We adopt a linear decay strategy for $\lambda$: $\lambda_t = \max(1 - \frac{t}{T_0}, 0)$, where $T_0$ is the preset maximum steps of the decay. Intuitively, a large $\lambda$ is desired at the beginning when each module is rarely optimized. Later, a small $\lambda$ is preferred to transit to normal training such that the forward inconsistency can be bridged. The remaining $T - T_0$ steps stick to normal training so that each quantized module adapts to its own predecessors.

Finally, an overview of the proposed parallel module-wise reconstruction error minimization with annealed teacher forcing is shown in Algorithm 1 and Algorithm 2. The Update$(\cdot)$ in Algorithm 2 can be any gradient update function such as AdamW [31] with learning rate $\eta^t$.

---

**Algorithm 1** Efficient PTQ for PLMs.

1: **procedure** *Main* ():
2:   Partition the PLM into $N$ modules.
3:   Fill in the input queues $\mathcal{I}, \hat{\mathcal{I}}$.
4:   **for** $n \leftarrow 1, ..., N$ **do**
5:     ▷ *run in parallel.*
6:     **while** $t < T$ **do**
7:       $\boldsymbol{f}_{l_{n-1}} \sim \mathcal{I}_{n-1}^t, \hat{\boldsymbol{f}}_{l_{n-1}} \sim \hat{\mathcal{I}}_{n-1}^t$.
8:       $\boldsymbol{f}_{l_n}^t, \hat{\boldsymbol{f}}_{l_n}^t \leftarrow$ MREM $(\boldsymbol{f}_{l_{n-1}}, \hat{\boldsymbol{f}}_{l_{n-1}}, t)$.
9:       Update $\mathcal{I}_n^t, \hat{\mathcal{I}}_n^t$ with $\boldsymbol{f}_{l_n}^t, \hat{\boldsymbol{f}}_{l_n}^t$.
10: **Return** the Quantized PLM.

---

**Algorithm 2** MREM algorithm.

1: **procedure** *MREM* $(\boldsymbol{f}_{l_{n-1}}, \hat{\boldsymbol{f}}_{l_{n-1}}, t)$:
2:   **if** $t < T_0$ **then**
3:     $\lambda_t \leftarrow \max(1 - \frac{t}{T_0}, 0)$.
4:     Compute $\tilde{\boldsymbol{f}}_{l_{n-1}}$ by Equation (4).
5:   Compute the full-precision module output $\boldsymbol{f}_{l_n}^t$.
6:   Compute the quantized module output $\hat{\boldsymbol{f}}_{l_n}^t$.
7:   Compute the loss $\ell^{(n)}$ by Equation (3).
8:   $\mathbf{w}_n^{t+1} \leftarrow$ Update$(\mathbf{w}_n^t, \frac{\partial \ell^{(n)}}{\partial \mathbf{w}_n^t}, \eta^t)$.
9:   $\mathbf{s}_n^{t+1} \leftarrow$ Update$(\mathbf{s}_n^t, \frac{\partial \ell^{(n)}}{\partial \mathbf{s}_n^t}, \eta^t)$.
10: **Return** $\boldsymbol{f}_{l_n}^t, \hat{\boldsymbol{f}}_{l_n}^t$.

---

# 4 Experiments

In this section, we empirically verify the proposed MREM for post-training quantization of PLMs. We first introduce the experimental setup in Section 4.1. Then we present main results in Section 4.2, including comparisons with QAT and REM, and other existing quantization baselines. In Section 4.4, we provide discussions on a variety of factors in our approach, such as the effect of teacher forcing, the number of model partitions, and calibration data size. Our implementation is based on MindSpore [1].

## 4.1 Experimental Setup

**Datasets and Metrics.** We evaluate post-training quantization on both the GLUE [45], and SQuAD benchmarks [39]. The size of calibration data is by default $|\tilde{\mathcal{D}}| = 4,096$, by randomly sampling instances from the full training set. As both RTE and MRPC tasks in the GLUE benchmark contain fewer than 4,096 samples, we use their full training set on these two tasks. We leave the study of data size in Section 4.4. Each experiment is repeated ten times with different calibration sets, and both the mean and standard deviations are reported.

We use the same evaluation metrics in [12, 56] for the development set of GLUE and SQuAD benchmarks. For results in Section 4.2, we report accuracies on both the matched section and mis-matched sections of MNLI, and EM (exact match) and F1 score for SQuAD. Additionally, we also report the training time (min), memory overhead (GB) as well as the size of the training set (K). We also provide comparisons with other existing methods in Section 4.3, where we adopt Matthews correlation for CoLA, Spearman correlation for STS-B, and accuracy for the rest ones (i.e., RTE, MRPC, SST-2, QQP, MNLI). We also report the averaged performance on GLUE as an overview.

**Implementation.** We use the standardly fine-tuned BERT-base and BERT-large models[3] on downstream tasks for both quantization-aware training and post-training quantization. We implement MREM in both the sequential training (abbv. MREM-S) in Section 3.1 and parallel training with teaching forcing (abbv. MREM-P) in Section 3.3. For each module, we train for $2,000$ steps with an initial learning rate of 1e-4 on GLUE tasks, and $4,000$ steps with an initial learning rate of 5e-5 on SQuAD datasets. The learning rate decays linearly as done in [24, 56]. By default, we partition the model into 4 modules on 4 NVIDIA-V100 GPUs. For baselines, we mainly compare with QAT and REM, where the former measures how far is PTQ from QAT, and the latter studies the effect of module-wise granularity in PTQ training. For a fair comparison of each method, we use the same quantization scheme for all methods, i.e., TWN [26] or LAQ [19] for 2-bit and 4-bit weight quantization, and LSQ [13] for all activation quantization. Unlike QAT that picks the best model based on the development set results, MREM is only tested once after training, which does not require further access to the development set. We leave more details of baseline implementation in Appendix B.1. The comparison with more published quantization baselines are left in Section 4.3.

## 4.2 Main Results: Comparison with QAT and REM

We first compare MREM-S and MREM-P with QAT and REM over MNLI and SQuAD benchmarks. We take BERT-base and BERT-large as backbone PLMs for quantization. The results on MNLI and SQuADv1.1 are summarized in Table 1 and Table 2 respectively, and results on SQuAD v2.0 are left to Table 6 in Appendix B.2. We again summarize according to the four dimensions in Section 2.2.

**Performance.** It can be found that our proposed MREM-S improves the performance of REM significantly given the same training time, and is much closer to QAT. For instance, according to in Table 1, MREM-S with 4-bit weight quantization on BERT-base and BERT-large achieves accuracies of $83.5\%_{\pm 0.1}$ and $86.1\%_{\pm 0.1}$ on the matched section of MNLI, which is on average $10.2\% \uparrow$ and $16.1\% \uparrow$ better than REM, and only $1.1\% \downarrow$ and $0.8\% \downarrow$ inferior to QAT, respectively. Moreover, with all modules trained in parallel, MREM-P is still close to or only slightly inferior to MREM-S. From Table 2, MREM-P can even outperform MREM-S with the "W2-E2-A4" quantized BERT-large model on SQuAD 1.1, i.e., the EM score and F1 score are on average $0.4\% \uparrow$ and $0.2\% \uparrow$ respectively.

**Training Time.** Our proposed MREM also enjoys significantly less training time than QAT. From Table 1, MREM only takes $84$ minutes to quantize BERT-large with 4-bit weights, which is about $38\times$

---

[3] We follow the default fine-tuning settings in Huggingface: https://github.com/huggingface/transformers.

Table 1: Results of our proposed MREM-S and MREM-P against QAT and REM on the development set of MNLI. "#Bits (W-E-A)" represents the bit-width for weights of Transformer layers, word embedding, and activations. Acc-m and Acc-mm denote accuracies on the matched and mismatched sections of MNLI respectively.

| #Bits (W-E-A) | Quant Method | BERT-base | | | | | BERT-large | | | | |
|---|---|---|---|---|---|---|---|---|---|---|---|
| | | Time (min)↓ | Mem (GB)↓ | # Data (K)↓ | Acc m(%)↑ | Acc mm(%)↑ | Time (min)↓ | Mem (GB)↓ | # Data (K)↓ | Acc m(%)↑ | Acc mm(%)↑ |
| *full-prec.* | N/A | 220 | 8.6 | 393 | 84.5 | 84.9 | 609 | 21.5 | 393 | 86.7 | 85.9 |
| 4-4-8 | QAT | $1,320$ | 11.9 | 393 | 84.6 | 84.9 | $3,180$ | 29.8 | 393 | 86.9 | 86.7 |
| | REM | 28 | 2.5 | 4 | $73.3_{\pm0.3}$ | $74.9_{\pm0.2}$ | 84 | 5.5 | 4 | $70.0_{\pm0.4}$ | $71.8_{\pm0.3}$ |
| | MREM-S | 36 | 4.6 | 4 | $83.5_{\pm0.1}$ | $83.9_{\pm0.1}$ | 84 | 10.8 | 4 | $86.1_{\pm0.1}$ | $85.9_{\pm0.1}$ |
| | MREM-P | 9 | $3.7_{\times4}$ | 4 | $83.4_{\pm0.1}$ | $83.7_{\pm0.1}$ | 21 | $8.6_{\times4}$ | 4 | $85.5_{\pm0.1}$ | $85.4_{\pm0.2}$ |
| 2-2-8 | QAT | 882 | 11.9 | 393 | 84.4 | 84.6 | $2,340$ | 29.8 | 393 | 86.5 | 86.1 |
| | REM | 24 | 2.5 | 4 | $71.6_{\pm0.4}$ | $73.4_{\pm0.4}$ | 64 | 5.5 | 4 | $66.9_{\pm0.4}$ | $68.6_{\pm0.7}$ |
| | MREM-S | 24 | 4.6 | 4 | $82.7_{\pm0.2}$ | $82.7_{\pm0.2}$ | 64 | 10.8 | 4 | $85.4_{\pm0.2}$ | $85.3_{\pm0.2}$ |
| | MREM-P | 6 | $3.7_{\times4}$ | 4 | $82.3_{\pm0.2}$ | $82.6_{\pm0.2}$ | 16 | $8.6_{\times4}$ | 4 | $84.6_{\pm0.2}$ | $84.6_{\pm0.1}$ |
| 2-2-4 | QAT | 875 | 11.9 | 393 | 83.5 | 84.2 | $2,280$ | 29.8 | 393 | 85.8 | 85.9 |
| | REM | 24 | 2.5 | 4 | $58.3_{\pm0.5}$ | $60.6_{\pm0.6}$ | 64 | 5.5 | 4 | $48.8_{\pm0.6}$ | $51.4_{\pm0.8}$ |
| | MREM-S | 24 | 4.6 | 4 | $81.1_{\pm0.2}$ | $81.5_{\pm0.2}$ | 64 | 10.8 | 4 | $83.6_{\pm0.2}$ | $83.7_{\pm0.2}$ |
| | MREM-P | 6 | $3.7_{\times4}$ | 4 | $80.8_{\pm0.2}$ | $81.2_{\pm0.2}$ | 16 | $8.6_{\times4}$ | 4 | $83.0_{\pm0.3}$ | $83.2_{\pm0.2}$ |

Table 2: Results of our proposed MREM-S and MREM-P against QAT and REM on the development set of SQuAD v1.1. "__" denotes results with two gradient accumulation steps under the same total batch size due to memory constraint.

| #Bits (W-E-A) | Quant Method | BERT-base | | | | | BERT-large | | | | |
|---|---|---|---|---|---|---|---|---|---|---|---|
| | | Time (min)↓ | Mem (GB)↓ | # Data (K)↓ | EM (%)↑ | F1 (%)↑ | Time (min)↓ | Mem (GB)↓ | # Data (K)↓ | EM (%)↑ | F1 (%)↑ |
| *full-prec.* | - | 177 | 11.7 | 88 | 81.5 | 88.7 | 488 | 30.4 | 88 | 86.9 | 93.1 |
| 4-4-8 | QAT | 428 | 18.4 | 88 | 80.2 | 87.9 | $\underline{1,920}$ | $\underline{27.0}$ | 88 | 86.7 | 93.0 |
| | REM | 65 | 3.1 | 4 | $46.1_{\pm0.5}$ | $60.0_{\pm0.5}$ | $\underline{175}$ | 7.3 | 4 | $68.3_{\pm0.1}$ | $79.3_{\pm0.1}$ |
| | MREM-S | 76 | 6.4 | 4 | $79.4_{\pm0.1}$ | $87.2_{\pm0.1}$ | 200 | 14.5 | 4 | $86.2_{\pm0.1}$ | $92.5_{\pm0.1}$ |
| | MREM-P | 19 | $5.5_{\times4}$ | 4 | $79.6_{\pm0.1}$ | $87.3_{\pm0.1}$ | 50 | $12.3_{\times4}$ | 4 | $86.0_{\pm0.1}$ | $92.4_{\pm0.1}$ |
| 2-2-8 | QAT | 335 | 18.4 | 88 | 79.3 | 87.2 | $\underline{1,200}$ | $\underline{27.0}$ | 88 | 86.1 | 92.5 |
| | REM | 60 | 3.1 | 4 | $40.1_{\pm0.4}$ | $55.0_{\pm0.4}$ | $\underline{160}$ | 7.3 | 4 | $66.4_{\pm0.5}$ | $77.7_{\pm0.3}$ |
| | MREM-S | 60 | 6.4 | 4 | $77.8_{\pm0.2}$ | $86.0_{\pm0.1}$ | 156 | 14.5 | 4 | $85.4_{\pm0.1}$ | $91.9_{\pm0.1}$ |
| | MREM-P | 15 | $5.5_{\times4}$ | 4 | $77.7_{\pm0.2}$ | $85.9_{\pm0.2}$ | 39 | $12.3_{\times4}$ | 4 | $85.3_{\pm0.2}$ | $91.8_{\pm0.1}$ |
| 2-2-4 | QAT | 331 | 18.4 | 88 | 77.1 | 85.9 | $\underline{1,186}$ | $\underline{27.0}$ | 88 | 84.7 | 93.1 |
| | REM | 60 | 3.1 | 4 | $10.4_{\pm0.2}$ | $24.6_{\pm0.2}$ | $\underline{160}$ | 7.3 | 4 | $28.3_{\pm0.6}$ | $45.0_{\pm0.5}$ |
| | MREM-S | 60 | 6.4 | 4 | $72.7_{\pm0.2}$ | $82.5_{\pm0.2}$ | 156 | 14.5 | 4 | $81.4_{\pm0.3}$ | $89.4_{\pm0.2}$ |
| | MREM-P | 15 | $5.5_{\times4}$ | 4 | $73.0_{\pm0.3}$ | $82.7_{\pm0.2}$ | 39 | $12.3_{\times4}$ | 4 | $81.8_{\pm0.3}$ | $89.6_{\pm0.2}$ |

faster than QAT and $7\times$ faster than full-precision fine-tuning. Note that the full-precision fine-tuning time is listed to conceptually compare how much time is further required for quantization. Comparing with REM, MREM does not cache the output after every matrix multiplication, which admits more iterations given the same amount of time. We discuss this further in Appendix B.3. Moreover, MREM-P is further **$4\times$** faster than MREM-S, which achieves the theoretical linear speedup on 4 GPUs. These together bring more than **$150\times$** reduction of training time when compared with QAT.

**Memory Overhead.** Our MREM-S and MREM-P take only around a third of the GPU memory by QAT, and a half of that by the full-precision fine-tuning, e.g., 29.8 GB for QAT and 10.8 GB for MREM-S on BERT-large. Moreover, we also encounter memory overflow (more than 32GB memory of an NVIDIA V100 GPU) for QAT training on SQuAD due to longer sequence length, and we adopt gradient accumulation that inevitably doubles the training time (i.e., underlined figures ("__") in Table 2). On the other hand, such issues can be easily mitigated in both REM and our proposed MREM, both of which can be even fed into a single NVIDIA GTX 1080 Ti GPU. We may further decrease the memory overhead of MREM to REM by increasing the number of modules, but this could harm the performance as discussed in Section 4.4.

**Data Accessibility.** Both REM and our proposed MREM follow the common practice of PTQ, relying on only $4,096$ randomly sampled instances on both MNLI and SQuAD, which is a tiny fraction of the original dataset. We shall provide more discussion on the calibration size in Section 4.4.

### 4.3 Main Results: Comparison with Existing Methods

Next, we compare our MREM with existing state-of-the-art BERT quantization methods. They include various QAT approaches such as Q-BERT [41], Quant-Noise [15], TernaryBERT [56], and

Table 3: Results on the GLUE development set. "Size" refers to model storage in "MB". "PTQ" indicates whether the method belongs to post-training quantization. "Avg." denotes the average results of all tasks.

| Quant Method | #Bits (W-E-A) | Size (MB) | PTQ | MNLI-m | QQP | QNLI | SST-2 | CoLA | STS-B | MRPC | RTE | Avg. |
|---|---|---|---|---|---|---|---|---|---|---|---|---|
| - | *full-prec.* | 418 | - | 84.9 | 91.4 | 92.1 | 93.2 | 59.7 | 90.1 | 86.3 | 72.2 | 83.9 |
| Q-BERT | 2-8-8 | 43 | ✗ | 76.6 | - | - | 84.6 | - | - | - | - | - |
| Q-BERT | 2/4-8-8 | 53 | ✗ | 83.5 | - | - | 92.6 | - | - | - | - | - |
| Quant-Noise | PQ | 38 | ✗ | 83.6 | - | - | - | - | - | - | - | - |
| TernaryBERT | 2-2-8 | 28 | ✗ | 83.3 | 90.1 | 91.1 | 92.8 | 55.7 | 87.9 | 87.5 | 72.9 | 82.7 |
| GOBO | 3-4-32 | 43 | ✓ | 83.7 | - | - | - | - | 88.3 | - | - | - |
| GOBO | 2-2-32 | 28 | ✓ | 71.0 | - | - | - | - | 82.7 | - | - | - |
| BRECQ | 8-4-4 | 61 | ✓ | 31.9 | 62.3 | 50.7 | 50.9 | 0.9 | 6.4 | 31.7 | 52.3 | 35.9 |
| QDrop | 8-4-4 | 61 | ✓ | 71.4 | 79.0 | 76.8 | 88.1 | 40.9 | 81.9 | 79.2 | 60.7 | 72.3 |
| REM | 4-4-8 | 28 | ✓ | $75.0_{\pm0.3}$ | $84.9_{\pm0.2}$ | $86.1_{\pm0.2}$ | $88.6_{\pm0.2}$ | $36.3_{\pm1.0}$ | $82.6_{\pm0.0}$ | $82.1_{\pm0.0}$ | $67.2_{\pm0.0}$ | $75.3_{\pm0.2}$ |
| REM | 2-2-8 | 28 | ✓ | $72.8_{\pm0.5}$ | $83.8_{\pm0.1}$ | $84.9_{\pm0.2}$ | $88.1_{\pm0.6}$ | $32.6_{\pm1.7}$ | $80.6_{\pm0.0}$ | $81.4_{\pm0.0}$ | $65.3_{\pm0.0}$ | $73.6_{\pm0.3}$ |
| REM | 2-2-4 | 28 | ✓ | $58.3_{\pm0.5}$ | $75.7_{\pm0.3}$ | $75.3_{\pm0.4}$ | $82.9_{\pm0.3}$ | $16.4_{\pm2.2}$ | $44.0_{\pm1.4}$ | $74.0_{\pm0.0}$ | $63.2_{\pm0.0}$ | $61.2_{\pm0.2}$ |
| MREM-S | 4-4-8 | 50 | ✓ | $83.5_{\pm0.1}$ | $90.2_{\pm0.1}$ | $91.2_{\pm0.1}$ | $91.4_{\pm0.4}$ | $55.1_{\pm0.8}$ | $89.1_{\pm0.1}$ | $84.8_{\pm0.0}$ | $71.8_{\pm0.0}$ | $82.4_{\pm0.1}$ |
|  | 2-2-8 | 28 | ✓ | $82.7_{\pm0.2}$ | $89.6_{\pm0.1}$ | $90.3_{\pm0.2}$ | $91.2_{\pm0.4}$ | $52.3_{\pm1.0}$ | $88.7_{\pm0.1}$ | $86.0_{\pm0.0}$ | $71.1_{\pm0.0}$ | $81.5_{\pm0.2}$ |
|  | 2-2-4 | 28 | ✓ | $81.1_{\pm0.1}$ | $88.7_{\pm0.1}$ | $89.4_{\pm0.1}$ | $90.8_{\pm0.3}$ | $46.4_{\pm1.1}$ | $87.7_{\pm0.1}$ | $83.8_{\pm0.0}$ | $70.8_{\pm0.0}$ | $80.0_{\pm0.1}$ |
| MREM-P | 4-4-8 | 50 | ✓ | $83.4_{\pm0.1}$ | $90.2_{\pm0.1}$ | $91.0_{\pm0.2}$ | $91.5_{\pm0.4}$ | $54.7_{\pm0.9}$ | $89.1_{\pm0.1}$ | $86.3_{\pm0.0}$ | $71.1_{\pm0.0}$ | $82.2_{\pm0.1}$ |
|  | 2-2-8 | 28 | ✓ | $82.3_{\pm0.2}$ | $89.4_{\pm0.1}$ | $90.3_{\pm0.2}$ | $91.3_{\pm0.4}$ | $52.9_{\pm1.2}$ | $88.3_{\pm0.2}$ | $85.8_{\pm0.0}$ | $72.9_{\pm0.0}$ | $81.6_{\pm0.2}$ |
|  | 2-2-4 | 28 | ✓ | $80.8_{\pm0.2}$ | $88.6_{\pm0.1}$ | $88.9_{\pm0.2}$ | $90.7_{\pm0.6}$ | $49.3_{\pm0.9}$ | $87.6_{\pm0.2}$ | $85.3_{\pm0.0}$ | $70.4_{\pm0.0}$ | $80.3_{\pm0.2}$ |

Table 4: Ablation studies of teacher forcing at different training steps over MNLI-m.

| #Bits (W-E-A) | # Steps | BERT-base | | BERT-large | |
|---|---|---|---|---|---|
|  |  | w/o TF | w/ TF | w/o TF | w/ TF |
| 2-2-8 | 250 | $79.6_{\pm0.3}$ | $80.7_{\pm0.2}$ | $82.1_{\pm0.4}$ | $83.1_{\pm0.2}$ |
|  | 500 | $81.0_{\pm0.3}$ | $81.6_{\pm0.3}$ | $83.4_{\pm0.3}$ | $84.1_{\pm0.3}$ |
|  | 2,000 | $82.2_{\pm0.2}$ | $82.7_{\pm0.2}$ | $84.3_{\pm0.3}$ | $84.6_{\pm0.2}$ |
|  | 4,000 | $82.3_{\pm0.3}$ | $82.5_{\pm0.2}$ | $84.5_{\pm0.2}$ | $84.7_{\pm0.2}$ |
| 2-2-4 | 250 | $73.9_{\pm0.5}$ | $77.3_{\pm0.4}$ | $76.5_{\pm0.9}$ | $79.3_{\pm0.4}$ |
|  | 500 | $77.9_{\pm0.2}$ | $79.0_{\pm0.2}$ | $80.0_{\pm0.5}$ | $81.4_{\pm0.2}$ |
|  | 2,000 | $80.4_{\pm0.2}$ | $80.8_{\pm0.2}$ | $82.5_{\pm0.4}$ | $83.0_{\pm0.3}$ |
|  | 4,000 | $80.7_{\pm0.2}$ | $81.0_{\pm0.2}$ | $83.1_{\pm0.1}$ | $83.3_{\pm0.3}$ |

Table 5: Comparison of BERT-base with and without per-channel quantization (PCQ) on MNLI.

| #Bits (W-E-A) | Methods | w/o PCQ | | w/ PCQ | |
|---|---|---|---|---|---|
|  |  | Acc m(%) | Acc mm(%) | Acc m(%) | Acc mm(%) |
| 4-4-8 | REM | $73.3_{\pm0.3}$ | $74.9_{\pm0.2}$ | $75.9_{\pm0.2}$ | $77.4_{\pm0.2}$ |
|  | MREM | $83.5_{\pm0.1}$ | $83.9_{\pm0.2}$ | $83.6_{\pm0.1}$ | $84.0_{\pm0.1}$ |
| 2-2-8 | REM | $71.6_{\pm0.4}$ | $73.4_{\pm0.4}$ | $74.1_{\pm0.5}$ | $75.6_{\pm0.5}$ |
|  | MREM | $82.7_{\pm0.2}$ | $82.7_{\pm0.2}$ | $82.8_{\pm0.1}$ | $82.9_{\pm0.1}$ |
| 2-2-4 | REM | $58.3_{\pm0.5}$ | $60.6_{\pm0.6}$ | $59.3_{\pm0.4}$ | $62.0_{\pm0.4}$ |
|  | MREM | $81.1_{\pm0.2}$ | $81.5_{\pm0.2}$ | $81.1_{\pm0.2}$ | $81.5_{\pm0.3}$ |

the PTQ baselines including GOBO [54], BRECQ [29], and QDrop [48]. The results are from the original papers. Additionally, we also compare our MREM with REM.

From Table 3, both our proposed MREM-S and MREM-P outperform existing PTQ approaches in most cases. For "W2-E2-A8" quantized models, MREM-S and MREM-P surpass GOBO by 11.7% ↑ and 11.3% ↑ on MNLI-m respectively. Both BRECQ and QDrop lag behind our approach even with the configuration "W4-E8-A4". For instance, the average scores on GLUE have the gaps of 9.3% and 8.0% between QDrop and our "W2-E2-A4" quantized MREM-S and MREM-P, respectively. Our results are even close to QAT approaches in multiple entries. For example, the "W4-E4-A8" quantized MREM-S and MREM-P have the mean accuracies of 83.5% and 83.4% on MNLI respectively, both of which are on par with the "W2/4-E8-A8" quantized Q-BERT.

### 4.4 Discussions

In this section, we provide further discussions to better understand the proposed approach. Unless specified otherwise, all experiments are built upon the BERT-base model over the MNLI dataset

**Teacher Forcing.** We now study how teacher forcing benefits MREM-P with different numbers of training steps, and results are listed in Table 4. We find that teacher forcing brings consistent improvement for both BERT-base and BERT-large, and the gain is more significant with fewer training steps or lower quantization bit-width. For example, it brings **3.4**% ↑ and **2.8**% ↑ on the "W2-E2-A4" quantized BERT-base and BERT-large respectively under 250 steps. This matches our intuition that fewer training steps or higher compression ratio give larger reconstruction error, when the clean input from the full-precision module can benefit more the quantized module. As more training steps bring only marginal improvement, we by default set the training steps to 2,000.

Additionally, we also plot training loss curves of the 2-nd and 4-th modules under 250 and 2,000 training steps in Figure 4, and more visualizations can be found in Figure 6 of Appendix B.4. We find that: 1) the loss curves with teacher forcing are apparently lower, especially when trained with fewer steps, which is consistent with Table 4; 2) the loss curves rise in the halfway during teacher forcing, since the modules need to adapt to quantized input with accumulated errors; 3) the loss curves of the 4-th modules are lower than the 2-nd ones, which matches the intuition that the late modules have

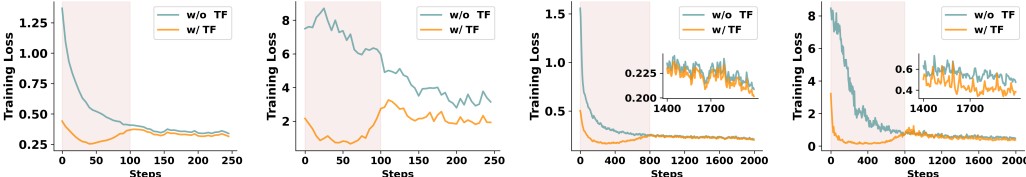

(a) Module-2 (250 Steps). (b) Module-4 (250 Steps). (c) Module-2 (2000 Steps).(d) Module-4 (2000 Steps).

Figure 4: The training loss curves with and without teacher forcing (TF) in MREM-P. The shaded area denotes teacher forcing in the first $40\%$ training steps. We show the 2-nd and 4-th modules trained with 250 steps and 2,000 steps in (a), (b) and (c), (d) respectively.

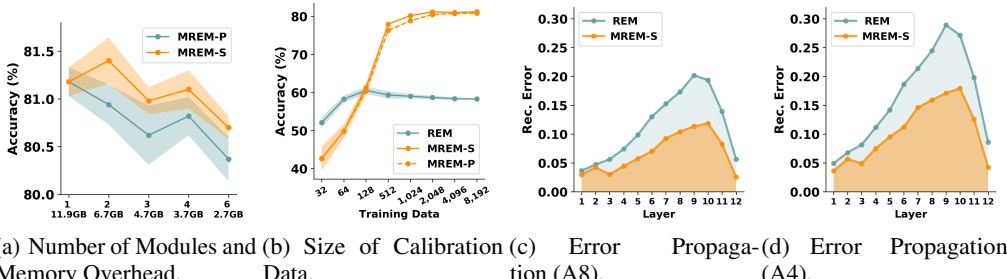

(a) Number of Modules and Memory Overhead. (b) Size of Calibration Data. (c) Error Propagation (A8). (d) Error Propagation (A4).

Figure 5: Discussions on the proposed MREM approach. In (a) and (b), the solid line and shaded area denote the averaged results and standard deviation of a "W2-E2-A4" quantized BERT-base model from 10 different seeds. (c) and (d) visualize the propagation of reconstruction error on "W2-E2-A8" and "W2-E2-A4" quantized BERT-base model, respectively.

more errors accumulated and thus benefit more from teacher forcing. We also try tuning the teacher forcing steps (i.e., shaded area) within $[20\%, 80\%]$ of total iterations, but observe no large difference in performance. We thus choose $40\%$ training steps for teacher forcing by default.

**Number of Modules and Memory Overhead.** We verify the effect of model partition on the final quantized performance, as well as their corresponding memory consumption. According to Figure 5(a), by varying the number of modules within $\{1, 2, 3, 4, 6\}$, it can be found that more model partitions give slightly lower performance, as layer-wise dependencies are less considered during reconstruction error minimization. However, more partitions lead to less running memory, i.e., $\{11.9, 6.7, 4.7, 3.7, 2.7\}$ GB for these partitions correspondingly. As the decrease of memory diminishes with more partitions, we partition the model into 4 modules by default.

**Size of Calibration Data.** The size of calibration data directly relates to the data accessibility issue in post-training quantization. To learn its effects, we vary the calibration data size $|\tilde{\mathcal{D}}|$ within $\{32, 64, 128, 512, 1024, 2048, 4096, 8192\}$, and list the results of REM, MREM-S and MREM-P. From Figure 5(b), it can be found that while REM is ahead of MREM-S/P with fewer than $128$ training samples, the accuracy of REM rises slowly and saturates at around $60\%$ afterwards. We hypothesize that the simple training objective in REM can hardly hold more training instances for optimization. MREM-S/P, on the other hand, can better exploit larger calibration data size, since the module-wise granularity admits higher flexibility for the optimization. As we find the diminishing gain to increase the training size after $4,096$ samples, we by default take $4,096$ samples.

**Reconstruction Error Propagation.** We visualize the propagation of reconstruction error for both "W2-E2-A8" and "W2-E2-A4" quantized BERT-base models in Figure 5(c) and Figure 5(d) respectively. It can be observed that our MREM achieves both lower values and slower rising rates of the reconstruction error than REM across all layers, which verifies the advantage of module-wise granularity to minimize the reconstruction error. Interestingly, while the reconstruction error generally gets enlarged layer-wisely in the first ten layers, it begins to decrease afterwards. We speculate this is due to the classification head that encourages concentrated hidden representations for the task.

**Per-channel Quantization.** Per-channel Quantization (PCQ) is prevalent in the post-training quantization of convolution neural networks [35, 33, 23]. To learn its effect in PLMs, PCQ assigns different quantization step-sizes at each output dimension of the linear layer, which is also known as row-wise quantization in [56]. The PCQ results of REM and MREM are shown in Table 5. It can be found that while PCQ improves REM by around $1.0\%$ to $2.5\%$, the gain is very incremental on MREM. Our results are also similar to the findings in [56], where the row-wise quantization brings little improvement. As PCQ requires to store more full-precision step sizes with minor improvement, we do not employ PCQ by default.

## 5 Related Work

Network quantization is a common approach to compress and accelerate deep neural networks [8, 26, 20, 19, 13, 28, 7]. For network quantization of pre-trained language models, previous efforts mostly adopt quantization-aware training [55, 41, 56, 3, 38]. For instance, Q8BERT [55] converts both parameters and activations with 8-bit representations with negligible task degradation. Q-BERT [41] exploits the hessian matrix of loss curvature to determine the layer-wise quantization bit-width, achieving a higher compression rate. TernaryBERT [56] proposes to ternarize BERT parameters with 2-bit representations, while BinaryBERT [3] further uses 1-bit to represent model parameters. BiBERT [38] explores the fully-binarized BERT with 1-bit weights and activations. Despite the success by these efforts, the heavy fine-tuning of QAT makes it prohibitive given constraints on training time, memory size, and data accessibility.

Post-training quantization [37, 53, 11] is thus considered mitigate these issues from QAT. One line of PTQ research quantizes the network purely without using any training data, but removes outliers in the full-precision parameters [57, 34, 16, 54]. Methodologically, PTQ can be achieved by splitting an outlier neuron with a large magnitude into two parts [57], where the magnitude can be halved. Alternatively, one can scale down outlier magnitude and multiply it back in subsequent layers, a.k.a weight equalization in [34]. Another solution is to treat the outliers and normal values in the distribution separately, by keeping two sets of quantization parameters [16, 54]. Another line in PTQ research [35, 47, 33, 23] minimizes the reconstruction error based on the calibration data. Compared with training-free PTQ approaches, such an approach significantly improves the performance of the quantized network, and we thus follow this line in this paper.

Recently, there are also concurrent efforts with us to apply PTQ for PTMs [49, 37, 53, 11]. In [49], a similar method is developed to minimize the quantization error via the calibration data. Moreover, PTQ is also explored for large generative PTMs (e.g. GPT-3 [5]) from 125M to 175B [37, 53, 11].

## 6 Conclusion

In this paper, we study post-training quantization for pre-trained language models. We show that existing quantization-aware training solutions suffer from slow training, huge memory overhead, and data privacy issues when accessing the full training set. To mitigate these issues, we propose module-wise reconstruction error minimization, an efficient solution to quantize PLMs. MREM can be conducted either sequentially or in parallel, where the parallel training can achieve the speedup close to the theoretical limit without apparent performance degradation. Experimental results show that the proposed solution greatly improves the performance. Meanwhile, it significantly reduces the training time and memory overhead with only thousands of training instances. We leave the broader impact and limitations of this work in Appendix D.

## Acknowledgement and Disclosure of Funding

We gratefully acknowledge the support of MindSpore for this research, as well as the insightful suggestions from the anonymous reviewers. The work described in this paper was partially supported by the National Key Research and Development Program of China (No. 2018AAA0100204), the Research Grants Council of the Hong Kong Special Administrative Region, China (No. CUHK 2410021 of the Research Impact Fund, No. R5034-18; and No. CUHK 14210920 of the General Research Fund).

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
