# OpenReview forum: "Towards Efficient Post-training Quantization of Pre-trained Language Models"
_NeurIPS.cc/2022/Conference — NeurIPS 2022 Accept_

### Official Review · Reviewer_Z1iD · 2022-07-09

**Rating:** 7
**Confidence:** 4
**Soundness:** 3 good
**Presentation:** 3 good
**Contribution:** 2 fair

**Summary:**

The paper proposes a technique named MREM to quantize a PLM without QAT, which requires huge resources (time, GPU memory, number of samples). Specifically, MREM adopts reconstruction error minimization in a module-wise manner. Each module consists of several Transformer layers, and each module is trained (almost) independent of the others by either taking full-precision output or the actual output from the previous module. To fully utilize such module-wise behavior, the paper also suggests dividing modules into each device. In addition, annealed teacher forcing is also used to simulate the error propagation through layers. The experiments demonstrate the savings (=advantages) of MREM for various resource types. Results of GLUE and SQuAD datasets show that MREM outperforms vanilla REM and reduces the gap between QAT and PTQ.

**Questions:**

The authors position MREM as a type of PTQ. However, I’m curious if it is okay to view the method that includes parameter update as a variant of PTQ. If this is a widely accepted classification, please ignore this question.

Can you compare the QAT case that only use the same number of (4K) samples? In that case, the resource usage gap will be much decreased, but how about the accuracy gap? Furthermore, if we allow using 4 GPUs to reduce the memory usage, can QAT also use multiple GPUs (reduce the batch size or employ model-parallel) for a fair comparison?


**Limitations:**

The authors addressed the limitations of this work.

**Strengths And Weaknesses:**

The paper is well written and easy to follow. Especially, the comparison between QAT and PTQ in Section 2.2 provides good motivation for the paper. The experiments are very well organized and support the advantages of the proposed method. Previous works are also sufficiently addressed.

Teacher forcing seems to be a good approach to dividing modules and performing separate optimization for each. The linear annealing schedule is reasonable, and the authors sufficiently support the necessity of the teacher forcing by experiments.

However, as the authors pointed out, the key idea that divides the network into several modules and applies reconstruction error minimization is already proposed in BRECQ [30]. There is some novelty in this work, but the improvements are somewhat incremental yet practical. Furthermore, there is no comparison between MREM and BRECQ in the Tables.

---

> ### Author Response · Authors · 2022-08-02
> **To Reviewer Z1iD (1/2)**
>
> We thank the reviewer for the insightful comments and suggestions.
>
> - **Q1**: "The key idea that divides the network into several modules and applies reconstruction error minimization is already proposed in BRECQ [30]. There is some novelty in this work, but the improvements are somewhat incremental yet practical. "
>
>   **A1**: Thank you for your recognition and this is a good question. While the idea to divide the network into modules (MREM-S) looks similar to BRECQ, it serves as the vanilla version of our approach and paves the way for the parallel training (MREM-P).  We would highlight that more innovations of this paper lie in MREM-P, which 1) takes advantage of the structure in split modules to achieve linear speed-up (e.g. 4x in 4 GPU) parallelly, and 2) the teacher forcing that further assists the parallel training as verified in the experiments. These techniques together can effectively reduce the PTQ process without much decrease in performance, which are not introduced in BRECQ.
>
> - **Q2**: "Furthermore, there is no comparison between MREM and BRECQ in the Tables."
>
>   **A2**: We list the public results of BRECQ [1] and its improved version QDrop [2] on GLUE below. The results are from the [QDrop paper](https://arxiv.org/pdf/2203.05740.pdf). Both BRECQ and QDrop are from the same group of researchers and their experiments are based on the same repository [GitHub - yhhhli/BRECQ: Pytorch implementation of BRECQ, ICLR 2021](https://github.com/yhhhli/BRECQ) . The "E8-W4-A4" quantized results of both BRECQ and QDrop are inferior to our "E2-W2-A4" results even with higher bits. While the overall idea of block-wise reconstruction error minimization is similar, there can be some potential differences in their implementation that result in worse performance: 1) they use 1K samples instead of 4K (but we can still outperform them on MNLI using 1K data according to Figure 5.b), and 2) they use for AdaRound [3] for weight quantization, which needs to carefully tune the annealing process for discretization. We have included these results in Appendix E of our revision.
>
>   |           | E-W-A Bits | RTE              | MRPC             | STS-B            | CoLA             | SST-2            | QNLI             | QQP              | MNLI-m           | Avg              |
>   | --------- | ---------- | ---------------- | ---------------- | ---------------- | ---------------- | ---------------- | ---------------- | ---------------- | ---------------- | ---------------- |
>   | BRECQ [1] | 8-4-4      | 52.3             | 31.7             | 6.39             | 0.9              | 50.9             | 50.7             | 62.3             | 31.9             | 35.9             |
>   | QDrop [2] | 8-4-4      | 60.7             | 79.2             | 81.9             | 40.9             | 88.1             | 76.8             | 79.0             | 71.4             | 72.3             |
>   | MREM-S    | 2-2-4      | $70.8_{\pm 0.0}$ | $83.8_{\pm 0.0}$ | $87.7_{\pm 0.1}$ | $46.4_{\pm 1.1}$ | $90.8_{\pm 0.3}$ | $89.4_{\pm 0.1}$ | $88.7_{\pm 0.1}$ | $81.1_{\pm 0.1}$ | $80.0_{\pm 0.1}$ |
>   | MREM-P    | 2-2-4      | $70.4_{\pm 0.0}$ | $85.3_{\pm 0.0}$ | $87.6_{\pm 0.2}$ | $49.3_{\pm 0.9}$ | $90.7_{\pm 0.4}$ | $88.9_{\pm 0.2}$ | $88.6_{\pm 0.1}$ | $80.8_{\pm 0.2}$ | $80.3_{\pm 0.2}$ |
>
> - **Q3**: "The authors position MREM as a type of PTQ. However, I’m curious if it is okay to view the method that includes parameter update as a variant of PTQ."
>
>   **A3**: The current literature on PTQ can be mainly divided into two strands of research. One line quantizes the network purely without using any training data, but simply removes outliers in the full-precision parameters [4, 5, 6]. The other line aims to minimize the reconstruction error using a very slight portion of unlabeled data (a.k.a. calibration set), which usually brings better performance without prolonging the quantization pipeline. Both our work, BRECQ [1] and QDrop [2] follow this line of research. We have further clarified these related work in Line 574 to Line 584 in our revision.
>
> - **Q4**: "Can you compare the QAT case that only use the same number of (4K) samples? In that case, the resource usage gap will be much decreased, but how about the accuracy gap?"
>
>   **A4**: Thanks for this insightful question. Training QAT with 4k samples reduces to a special case of our proposed MREM-S/P, i.e., #modules=1, without parallel-training and teacher-forcing. According to Figure 5.a: the performance of MREM-S/P with 1-module (i.e., QAT) is $81.2_{\pm 0.1}$, while MREM-S/P with 4 modules (by default) are $81.1_{\pm 0.2}$ and $80.8_{\pm 0.2}$ respectively.  However, we remark that even though QAT training with 4K data achieves slightly better performance, this comes at the cost of increasing memory and longer training time. We also provide more experiments on this point in the reply to Q5 below.

---

> > ### Author Response · Authors · 2022-08-02
> > **To Reviewer Z1iD (2/2)**
> >
> > - **Q5**: "Furthermore, if we allow using 4 GPUs to reduce the memory usage, can QAT also use multiple GPUs (reduce the batch size or employ model-parallel) for a fair comparison?"
> >
> >   **A5**:
> >   **QAT with data parallelism:** We implement the data-parallel training with `torch.nn.parallel.DistributedDataParallel` for QAT. We keep all configurations consistent with MREM-P (e.g., 4K training samples, and learning rate as 5e-5), except for reducing the batch-size and training steps to keep the same memory consumption/training time. The results of BERT-base over SQuAD 1.1 are listed below. It can be found that:
> >
> >   1) **Under the same memory consumption**: As QAT requires to load the entire model into the GPU memory, it can only hold 1 sample per GPU match the memory of MREM-P, which leads to worse performance;
> >
> >   2) **Under the same batch size:** Even when the batch size of QAT are consistent with MREM-P (though it consumes the additional 3.7GB memory), the performance is still not as good as MREM-P since the full back-propagation is more time-consuming and thus allows fewer training iterations within 15 minutes.
> >
> >   | E-W-A Bits | Method | Total Bsz | Mem/GPU (GB) | # Iters | Time (min) | EM               | F1               |
> >   | ---------- | ------ | --------- | ------------ | ------- | ---------- | ---------------- | ---------------- |
> >   | 4-4-8      | QAT    | 4 (1x4)   | 6.4          | 3000    | 15.5       | $74.7_{\pm 0.2}$ | $83.4_{\pm 0.1}$ |
> >   |            | QAT    | 12 (3x4)  | 9.2          | 2600    | 15         | $77.9_{\pm 0.1}$ | $85.7_{\pm 0.1}$ |
> >   |            | MREM-P | 12        | 5.5          | 4000    | 15         | $79.6_{\pm 0.1}$ | $87.3_{\pm 0.1}$ |
> >   | 2-2-8      | QAT    | 4 (1x4)   | 6.4          | 3000    | 15.5       | $71.5_{\pm 0.3}$ | $80.9_{\pm 0.2}$ |
> >   |            | QAT    | 12 (3x4)  | 9.2          | 2600    | 15         | $74.4_{\pm 0.3}$ | $83.2_{\pm 0.2}$ |
> >   |            | MREM-P | 12        | 5.5          | 4000    | 15         | $77.7_{\pm 0.2}$ | $85.9_{\pm 0.2}$ |
> >   | 2-2-4      | QAT    | 4 (1x4)   | 6.4          | 3000    | 15.5       | $65.9_{\pm 0.4}$ | $76.6_{\pm 0.3}$ |
> >   |            | QAT    | 12 (3x4)  | 9.2          | 2600    | 15         | $69.4_{\pm 0.3}$ | $79.4_{\pm 0.2}$ |
> >   |            | MREM-P | 12        | 5.5          | 4000    | 15         | $73.0_{\pm 0.3}$ | $82.7_{\pm 0.2}$ |
> >
> >   **QAT with model parallelism:** Due to limited time it's hard to implement the pipeline parallel (e.g., GPipe) for QAT in a hurry. Yet we have provided detailed analysis in Appendix B (Line 621) that explains the difference of our MREM-P with pipeline parallel. When applying pipeline parallel for QAT, it suffers from bubble time that scales with the model partitions N or micro-batches M.  On the contrary, our MREM-P has negligible bubble time and thus can achieve nearly theoretical speed-up (i.e., 4x on 4 GPUs). Yet we still need to point out the potential drawback of MREM-P: despite the empirical success in the PTQ setting, MREM-P is still local training with blocked gradients between adjacent modules (Line 676). Thus QAT with pipeline parallel may outperform MREM-P when longer training time is allowed or more training samples are available.
> >
> > [1] Li, et al. Brecq: Pushing the limit of post-training quantization by block reconstruction. ICLR, 2021.
> >
> > [2] Wei, et al. QDrop: Randomly Dropping Quantization for Extremely Low-bit Post-Training Quantization. ICLR 2022.
> >
> > [3] Nagel, et al. Up or Down? Adaptive Rounding for Post-Training Quantization. ICML 2020.
> >
> > [4] Zhao, et al. Improving neural network quantization without retraining using outlier channel splitting. ICML 2019.
> >
> > [5] Nagel, et al. Data-free quantization through weight equalization and bias correction. CVPR 2019.
> >
> > [6] Zadeh, et al. Gobo: Quantizing attention-based nlp models for low latency and energy efficient inference. Preprint arXiv, 2020.

---

> > > ### Comment · Reviewer_Z1iD · 2022-08-04
> > > **Thank you for the authors' response**
> > >
> > > Thank you for your effort in additional experiments.
> > > Especially, the experiment in A5) presents very interesting results, and also shows the advantage of the proposed MREM.
> > > I can agree that MREM is a variant of PTQ that only utilizes a few calibration samples that not only decide the step size but also modify the parameter itself.

---

### Official Review · Reviewer_JA5e · 2022-07-11

**Rating:** 6
**Confidence:** 3
**Soundness:** 3 good
**Presentation:** 3 good
**Contribution:** 3 good

**Summary:**

The paper proposes an approach for post-training quantization of pre-trained language models. The core of the idea is to run the unquantized and quantized networks in parallel for a small calibration dataset, and update the quantized network parameters such that the quantized network activations match the unquantized reference activations at certain module boundaries in the network. Similar to teacher forcing, the accumulated error in the activations as a result of quantization may be periodically reset/reduced, allowing better training in a regime where total accumulated error would otherwise be too large. Additionally, the model may be spread across multiple GPUs during quantization, and queues may be introduced to allow the GPUs to run asynchronously without waiting for stragglers.

The method results in quantized model accuracy that is noticeably higher than a baseline for post-training quantization. Compared to quantization-aware training, the proposed method is faster, avoids memory pressure by using less memory, and has better parallelization (though accuracy for the proposed method doesn't quite match QAT).

**Questions:**

Could the authors please explain when quantization with respect to being before/after/joint-with task-specific fine-tuning?

My initial assumption would be that PQT occurs after task-specific fine-tuning (though I don't think this is ever stated explicitly in the paper, so I could be wrong). However, Tables 1&2 include a "full-precision" row alongside the quantization approaches, and MREM uses less time/memory/data than full-precision fine-tuning. These numbers wouldn't be measuring the same quantity if MREM is run only after full-precision fine-tuning has already occurred. I am further confused by what "time" means in the QAT context. With QAT it's technically possible to run the entire task-specific fine-tuning with QAT enabled, or to first fine-tune a bit in the standard full-precision way and then add QAT and optimize the task loss some more. It's again not clear to me what is being done, and what columns like "time" actually represent.

**Limitations:**

Yes

**Strengths And Weaknesses:**

A strength of the method is its performance across the four key axes identified in the paper: training time, memory overhead, data requirements, and accuracy. The paper provides a clear description of the base method and how to parallelize it.

In terms of the weaknesses of the paper, I would appreciate the authors' clarification regarding the Questions section below; as it stands I have some confusion on how to interpret the results.

The paper also presents REM as the most suitable baseline for PTQ, but the accuracy results (Table 3) instead feature GOBO and no REM. It would be better to have more thorough experiments or exposition instead of this jump. It would also be helpful to have a PQT baseline in Table 3 that runs on more than just two GLUE tasks, because this may be too few to draw good comparisons of MREM accuracy vs. other PQT approaches.

Minor:
- superscript formatting for 1e-4/5e-5 on lines 206-207 is awkward to read (e^x usually means exp(x))
- 103: misspelled calibration

---

> ### Author Response · Authors · 2022-08-02
> **To Reivewer JA5e**
>
> We thank the reviewer for your insightful comments.
>
> - **Q1**: "Could the authors please explain when quantization with respect to being before/after/joint-with task-specific fine-tuning? ... My initial assumption would be that PQT occurs after task-specific fine-tuning (though I don't think this is ever stated explicitly in the paper, so I could be wrong). ... I am further confused by what "time" means in the QAT context. ... It's again not clear to me what is being done, and what columns like"time" actually represent."
>
>   **A1**: The process for QAT/PTQ is as follows: We first task-specifically fine-tune the pretrained BERT in full-precision, and then start QAT/PTQ (as mentioned in L203), which is a common approach that aligns with previous BERT quantization papers [1,2,3]. In Table 1&2, we list both the training time of full-precision fine-tuning and QAT/PTQ. Our primary goal is to show how much time is further required for quantization after full-precision fine-tuning, where QAT takes much longer time than fine-tuning while PTQ does not need much time additionally.
>   We have clarified this in the paper revision (Line 204, Line 232 and caption of Table 1).
>
> - **Q2**: "The paper also presents REM as the most suitable baseline for PTQ, but the accuracy results (Table 3) instead feature GOBO and no REM. It would be better to have more thorough experiments or exposition instead of this jump. It would also be helpful to have a PQT baseline in Table 3 that runs on more than just two GLUE tasks,because this may be too few to draw good comparisons of MREM accuracy vs. other PQT approaches."
>
>   **A2**: Thanks for your suggestion. For REM, we keep the same experimental setup with MREM-S/P, and its results on all GLUE tasks are listed below. It can be found that REM is inferior to both MREM-S and MREM-P on each GLUE task. We have included these results in Appendix E of our revision.
>
>   |        | E-W-A Bits | RTE              | MRPC             | STS-B            | CoLA             | SST-2            | QNLI             | QQP              | MNLI-m           | Avg              |
>   | ------ | ---------- | ---------------- | ---------------- | ---------------- | ---------------- | ---------------- | ---------------- | ---------------- | ---------------- | ---------------- |
>   | REM    | 4-4-8      | $67.2_{\pm 0.0}$ | $82.1_{\pm 0.0}$ | $82.6_{\pm 0.0}$ | $36.3_{\pm 1.0}$ | $88.6_{\pm 0.2}$ | $86.1_{\pm 0.2}$ | $84.9_{\pm 0.1}$ | $75.0_{\pm 0.3}$ | $75.3_{\pm 0.1}$ |
>   | MREM-S | 4-4-8      | $71.8_{\pm 0.0}$ | $84.8_{\pm 0.0}$ | $89.1_{\pm 0.1}$ | $55.1_{\pm 0.8}$ | $91.4_{\pm 0.4}$ | $91.2_{\pm 0.1}$ | $90.2_{\pm 0.1}$ | $83.5_{\pm 0.1}$ | $82.4_{\pm 0.1}$ |
>   | MREM-P | 4-4-8      | $71.1_{\pm 0.0}$ | $86.0_{\pm 0.0}$ | $88.7_{\pm 0.1}$ | $52.3_{\pm 1.0}$ | $91.2_{\pm 0.4}$ | $90.3_{\pm 0.2}$ | $89.6_{\pm 0.1}$ | $82.7_{\pm 0.2}$ | $81.5_{\pm 0.2}$ |
>   | REM    | 2-2-8      | $65.3_{\pm 0.0}$ | $81.4_{\pm 0.0}$ | $80.6_{\pm 0.0}$ | $32.6_{\pm 1.7}$ | $88.1_{\pm 0.6}$ | $84.9_{\pm 0.2}$ | $83.8_{\pm 0.1}$ | $72.8_{\pm 0.5}$ | $73.6_{\pm 0.3}$ |
>   | MREM-S | 2-2-8      | $71.1_{\pm 0.0}$ | $86.3_{\pm 0.0}$ | $89.1_{\pm 0.1}$ | $54.7_{\pm 0.9}$ | $91.5_{\pm 0.4}$ | $91.0_{\pm 0.2}$ | $90.2_{\pm 0.1}$ | $83.4_{\pm 0.1}$ | $82.2_{\pm 0.1}$ |
>   | MREM-P | 2-2-8      | $72.9_{\pm 0.0}$ | $85.8_{\pm 0.0}$ | $88.3_{\pm 0.2}$ | $52.9_{\pm 1.2}$ | $91.3_{\pm 0.4}$ | $90.3_{\pm 0.2}$ | $89.4_{\pm 0.1}$ | $82.3_{\pm 0.2}$ | $81.6_{\pm 0.2}$ |
>   | REM    | 2-2-4      | $63.2_{\pm 0.0}$ | $74.0_{\pm 0.0}$ | $44.0_{\pm 1.4}$ | $16.4_{\pm 2.2}$ | $82.9_{\pm 0.3}$ | $75.3_{\pm 0.4}$ | $75.7_{\pm 0.3}$ | $58.3_{\pm 0.5}$ | $61.2_{\pm 0.2}$ |
>   | MREM-S | 2-2-4      | $70.8_{\pm 0.0}$ | $83.8_{\pm 0.0}$ | $87.7_{\pm 0.1}$ | $46.4_{\pm 1.1}$ | $90.8_{\pm 0.3}$ | $89.4_{\pm 0.1}$ | $88.7_{\pm 0.1}$ | $81.1_{\pm 0.1}$ | $80.0_{\pm 0.1}$ |
>   | MREM-P | 2-2-4      | $70.4_{\pm 0.0}$ | $85.3_{\pm 0.0}$ | $87.6_{\pm 0.2}$ | $49.3_{\pm 0.9}$ | $90.7_{\pm 0.4}$ | $88.9_{\pm 0.2}$ | $88.6_{\pm 0.1}$ | $80.8_{\pm 0.2}$ | $80.3_{\pm 0.2}$ |
>
>   Additionally, we also provide the PTQ results of all GLUE tasks from BRECQ[4] and QDrop[5]. Please refer to our reply to Q2 of Reviewer Z1iD. It can be found that our MREM-S and MREM-P still outperform their "E8-W4-A4" configuration on all GLUE tasks by a large margin.
>
> - Minor updates: thanks for your correction and we have revised them accordingly in the paper.
>
> [1] Zhang, et al. Ternarybert: Distillation-aware ultra-low bit bert. EMNLP 2020.
>
> [2] Bai, et.al. Binarybert: Pushing the limit of bert quantization. ACL 2021.
>
> [3] Shen, et al. Q-bert: Hessian based ultra low precision quantization of bert. AAAI 2020.
>
> [4] Li, et al. Brecq: Pushing the limit of post-training quantization by block reconstruction. ICLR, 2021.
>
> [5] Wei, et al. QDrop: Randomly Dropping Quantization for Extremely Low-bit Post-Training Quantization. ICLR 2022.

---

### Official Review · Reviewer_zLnQ · 2022-07-11

**Rating:** 5
**Confidence:** 4
**Soundness:** 3 good
**Presentation:** 2 fair
**Contribution:** 2 fair

**Summary:**

The paper presented another way to perform post train quantization (PTQ) on BERT models with a parallel optimization method to speed up the fine-tuning process.

**Questions:**

As listed under "Weaknesses".

**Limitations:**

This paper provides a decent discussion of the limitation.

**Strengths And Weaknesses:**

Strength:
1. The paper offers a detailed ablation study on each of the auxiliary training methodologies they are using on top of the main PTQ algorithm.
2. The paper provides extensive experimental results covering previous SOTA methods in the field of PTQ and QAT.
3. the paper provides a detailed explanation of the methodologies and background, helping the understanding of the paper.

Weaknesses:
1. This work provides very few intuitive/theoretical explanations of why their method could sometimes work as well as QAT methods. In particular, since the paper adapts the well-known quantization function Q(w) = s*round(clip(w/s, MAX, MIN)) (I assume this paper uses the classical form of this quantization function as the author failed to provide explicit details), which is originally a QAT method, it would be ideal that (a) some comparison of loss, accuracy, reconstruction errors, etc. of QAT and/or (b) the STE optimization results over the calibration set is presented. So far, it is hard to see which part of the method is contributing to the most accuracy.
2. This work seems to be heavily based on _Learned Step Size Quantization_, a well-known work in the field of quantization. It seems to me that the only work this paper made is directly adapting this method to the field of PTQ and showing that it is working well with both limited data (PTQ calibration) and module-wise REM. Due to the lack of ablation study in point 1, the novelty seems to be limited as this paper failed to present how significant their optimization methods could bring benefits to LSQ.

---

> ### Author Response · Authors · 2022-08-02
> **To Reviewer zLnQ (1/3)**
>
> Thank for the reviewer's comments.
>
> - **Q1**: "This work provides very few intuitive/theoretical explanations of why their method could sometimes work as well as QAT methods."
>
>   **A1**: We answer this question from the following two aspects:
>
>   1. **PTQ has low reconstruction errors close to QAT**: The proposed PTQ approaches in this work share the similar training objective as QAT methods like TernaryBERT [1]: mimicking the full-precision model by minimizing the reconstruction error of quantization, but with significantly less training data and time. Below we show the minimized reconstruction error (i.e., $\ell_2$ norm of the hidden states at the last Transformer layer  as defined in Eq. (2)) of quantized BERT-base on MNLI for QAT, REM, MREM-S, and MREM-P. It can be found that a smaller reconstruction error corresponds to a higher accuracy, which matches our intuition. Our MREM-S/P has close reconstruction error to QAT, and this explains its good performance.
>
>   | E-W-A | Method | Rec Error | Acc-m  |     | E-W-A | Method | Rec Error | Acc-m  |     | E-W-A | Method | Rec Error | Acc-m  |
>   | ----- | ------ | --------- | ------ | --- | ----- | ------ | --------- | ------ | --- | ----- | ------ | --------- | ------ |
>   | 4-4-8 | QAT    | 1.4e-2    | $84.6$ |     | 2-2-8 | QAT    | 1.9e-2    | $84.4$ |     | 2-2-4 | QAT    | 3.1e-2    | $83.5$ |
>   |       | REM    | 5.2e-2    | $73.3$ |     |       | REM    | 5.6e-2    | $71.6$ |     |       | REM    | 8.6e-2    | $58.3$ |
>   |       | MERM-S | 1.9e-2    | $83.5$ |     |       | MERM-S | 2.5e-2    | $82.7$ |     |       | MERM-S | 4.3e-2    | $81.1$ |
>   |       | MREM-P | 2.2e-2    | $83.4$ |     |       | MREM-P | 2.6e-2    | $82.3$ |     |       | MREM-P | 4.8e-2    | $80.8$ |
>
>   2. **Why does PTQ work? Sample efficient module-wise training**: As is discussed at Line 96, one important factor to explain the success of PTQ is the sample-efficiency of the layer/module-wise training in Eq. (2) and Eq. (3). As is theoretically explained in [2], the discrepancy between two models (a.k.a. reconstruction error in our case) depends linearly on the number of model partitions $N$ for module-wise training. However, it scales double exponentially with $N$ for end-to-end training in QAT. In fact, similar ideas (greedy layer/module-wise training over calibration data) has been commonly applied in PTQ for computer vision models [3,4,5,6], and their performances are quite  close to QAT as well (e.g, the W4-A4 PTQ quantized ResNet-18 with BRECQ [3] is only 1.5% inferior to LSQ on ImageNet).
>
> - **Q2**: "In particular, since the paper adapts the well-known quantization function Q(w) =s*round(clip(w/s, MAX, MIN)) (I assume this paper uses the classical form of this quantization function as the author failed to provide explicit details), which is originally a QAT method."
>
>   **A2**: Equation 1 is a general quantization function that can be integrated both in QAT or PTQ. Both weight and activation quantization methods used in this paper fall into this definition. Specifically, we use TWN/LAQ for weight quantization, and LSQ for activation quantization (as mentioned in L211) to fairly compare all methods in Table 1&2. We have include more details on how these quantization functions relate to Equation 1 in Appendix F of our revised version.
>
>   We highlight again that the key challenges to address in PTQ is to quantize networks under harsh constraints of data accessibility, training time, and memory overhead (as listed in Section 2.2), rather than designing a new quantization function. Instead, existing quantization functions (including LSQ) are widely used in various PTQ approaches [3,5,6,7].
>
> - **Q3**: "It would be ideal that (a) some comparison of loss, accuracy, reconstruction errors, etc. of QAT and/or (b) the STE optimization results over the calibration set is presented. So far, it is hard to see which part of the method is contributing to the most accuracy."
>
>   **A3**:
>
>   1. a) The comparisons of the reconstruction error (which is also the loss objective) and the associated accuracies are presented in A1 above.
>      b) While we are not sure about what the reviewer mean by "STE optimization results", we use the STE with identity mapping for weight quantization (TWN and LAQ), and strictly follow the STE rule in LSQ to update both model weights and step sizes. These hold the same for QAT, REM and our proposed MREM.
>    2. The key factors contributing to the good performance of MREM comes from the proposed module-wise reconstruction error minimization  as well as the parallel training strategy with teacher forcing, and each of them are ablated in Section 4.4.

---

> > ### Author Response · Authors · 2022-08-02
> > **To Reviewer zLnQ (2/3)**
> >
> > - **Q4**: "Due to the lack of ablation study in point 1, the novelty seems to be limited as this paper failed to present how significant their optimization methods could bring benefits to LSQ. ... .This work seems to be heavily based on Learned Step Size Quantization. Using LSQ (with some modification) is not really new in BERT."
> >
> >   **A4**:
> >
> >   1. As mentioned above in our reply to Q2, all methods (QAT, REM and our MREM-S/P) in Table 1&2 use the same TWN/LAQ for weight quantization and LSQ for activation quantization for fair comparisons. This indicates that the proposed MREM-S/P are agnostic to the choice of quantization functions.
> >
> >   2. In addition, we also perform another experiment that replaces the original LSQ with two other quantization methods (i.e., classical symmetric uniform quantization and PACT [8]). For PACT, we use the same learning rate (1e-4) of step-size in LSQ to update the clipping boundary. We keep all the rest setup consistent with Table 1, and use MREM-P over BERT-base for illustration. From the results below, both symmetric uniform quantization and PACT still achieve competitive results with LSQ even without careful tuning of hyper-parameters. This again affirms that the proposed MREM-S/P is a general method that can be combined with various quantization functions.
> >
> >   | E-W-A | Act Quant | Acc-m            |     | E-W-A | Act Quant | Acc-m            |     | E-W-A | Act Quant | Acc-m            |
> >   | ----- | --------- | ---------------- | --- | ----- | --------- | ---------------- | --- | ----- | --------- | ---------------- |
> >   | 4-4-8 | Uniform   | $83.3_{\pm 0.1}$ |     | 2-2-8 | Uniform   | $82.1_{\pm 0.2}$ |     | 2-2-4 | Uniform   | $80.1_{\pm 0.1}$ |
> >   |       | PACT      | $83.1_{\pm 0.1}$ |     |       | PACT      | $81.7_{\pm 0.3}$ |     |       | PACT      | $80.9_{\pm 0.2}$ |
> >   |       | LSQ       | $83.4_{\pm 0.1}$ |     |       | LSQ       | $82.3_{\pm 0.2}$ |     |       | LSQ       | $80.8_{\pm 0.2}$ |
> >
> > - **Q5**: "MKQ-BERT, KDLSQ-BERT...Both works show that there can be large performance degradations in multiple tasks under extreme compression. In particular, KDLSQ-BERT reported 2-2-8 LSQ QAT results on CoLA and RTE that are even (significantly) lower than the numbers reported in this paper. I am concerned that the authors had accidentally made some mistakes in the experiment. Ideally, the authors could provide some study to show why this is the case."
> >
> >   **A5**:  We have provided our code in the supplementary materials. The results reported in the paper can be reproduced following instructions in `readme.md`. We will also release our code upon acceptance. As for the reason behind the poor performance of MKQ-BERT and KDLSQ-BERT, since their code are not publically available, we are not aware of their implementation details.

---

> > > ### Author Response · Authors · 2022-08-02
> > > **To Reviewer zLnQ (3/3)**
> > >
> > > - **Q6**: "This paper failed to provide some details (which can be quite important) of the experiment and may hinder reproducing the results listed in the paper. Since the step-size quantizer is used, the paper should cover at least (a) how they are initialized; (b) if there is additional gradient scaling to step-size (since this is mentioned as a crucial step in the LSQ paper); (c) to what extend the step-size parameters are shared, is it per layer, per module, or some other settings."
> > >
> > >   **A6**: When using LSQ for activation quantization, we strictly follow the implementation in the original paper:
> > >
> > >   a) the step-sizes of activations $v$ are initialized to $2 |v| / \sqrt Q_P$ based on the first batch of calibration data;
> > >
> > >   b) we scale the gradients by $1/\sqrt{N_F Q_P}$ for activations with $N_F$ elements and $Q_P$ quantization points;
> > >
> > >   c) The step-size is shared per-layer. We have added these implementation details in Appendix F of the revised paper. You may also refer to our code (`class SymLsqQuantizer` in `transformer/utils_quant.py`) for details.
> > >
> > > - **Q7**: "But I am not 100% sure what the projection function \Pi(.) is, while it is mentioned that this quantization function is "symmetric uniform," which most likely means round(clip(.)), this should be made explicit as this is one of the most crucial equations in the paper."
> > >
> > >   **A7**: Eq. (1) and the description of it in L80-81 have already mathematically defined the projection function $\Pi(\cdot)$ explicitly, i.e., the projection function maps $x/s$ to its closest integer in $\Omega(b)=\{-2^{b-1}+1, ...,0,...,2^{b-1}-1\}$.
> > >
> > > [1] Zhang, et al. TernaryBERT: Distillation-aware Ultra-low Bit BERT. EMNLP 2020.
> > >
> > > [2] Zhou, et al. Go wide, then narrow: Efficient training of deep thin networks. ICML 2020.
> > >
> > > [3] Li, et al. Brecq: Pushing the limit of post-training quantization by block reconstruction. ICLR, 2021.
> > >
> > > [4] Nagel, et al. Up or Down? Adaptive Rounding for Post-Training Quantization. ICML 2020.
> > >
> > > [5] Hubara, et al. Improving Post Training Neural Quantization: Layer-wise Calibration and Integer Programming. NeurIPS 2021.
> > >
> > > [6] Wei, et al. QDrop: Randomly Dropping Quantization for Extremely Low-bit Post-Training Quantization. ICLR 2022.
> > >
> > > [7] Nagel, et al. Data-free quantization through weight equalization and bias correction. CVPR 2019.
> > >
> > > [8] Choi J, et al. Pact: Parameterized clipping activation for quantized neural networks. arXiv preprint arXiv:1805.06085, 2018.

---

> > > > ### Comment · Reviewer_zLnQ · 2022-08-09
> > > > **Thank you for the authors' response**
> > > >
> > > > I really appreciate the author's effort in the extra experiments and extra explanations. I would say that they addressed most of my concerns 3,4. For 1 (in summary, comparison with well-known QAT method), I appreciate that the author includes more comparison with methodologies such as BRECQ -- this would provide stronger results in showing the effectiveness of MERM and raise my score. But the key I am looking for is one more error analysis w.r.t. some well-known QAT method. As the authors mentioned MERM performs better than the baselines because it significantly reduces the error propagation. This makes sense in explaining that MERM is **better** (which I strongly agree with), but still, I cannot tell **what is a good propagation error**, from fig 5c/5d. From my perspective, MERM still seems to incur quite a significant amount of error. Thus, this point is only partially addressed. Alternatively, this could also be addressed by citing previous work that does this study.
> > > >
> > > > As for point 2, I appreciate the author making the difference clear -- originally I was confused -- and the new appendix is quite useful in understanding how MERM is implemented in detail. The additional experiment (uniform - PACT - LSQ table) the author provided does present a better clarification than simply stacking the QAT methods and using them in the PTQ field. However, as my original concern is, the quantizer this paper used is adapted from QAT fields, and thus it is better to present comparisons (like error analysis in Fig 5c/5d) to some QAT methods in the appendix. This would provide robust justifications (both result and reasoning) for an alternative approach that is almost as good as QAT but saves tons of resources - this would make the method provided in this paper way stronger.
> > > >
> > > > In summary, the concerns I still have: no error analysis w.r.t. some QAT baseline, which I believe is needed given the context of this paper. But given the additional information provided, I raised my score to 5.

---

> > > > > ### Author Response · Authors · 2022-08-10
> > > > > **More discussions on the reconstruction error**
> > > > >
> > > > > We really appreciate the reviewer's response. The reconstruction error of the last transformer layer is already shown in the A1 to Q1.
> > > > >
> > > > > To further clarify this point, we also show the layer-wise reconstruction error of QAT, REM, MREM-S and MREM-P below. The results are based on the W2-E2-A8 quantized BERT-base on MNLI (i.e., the same configuration to Fig 5c.)
> > > > >
> > > > > | Method | L1  | L2  | L3  | L4  | L5  | L6  | L7  | L8  | L9  | L10 | L11 | L12 |
> > > > > | --- | --- | --- | --- | --- | --- | --- | --- | --- | --- | --- | --- | --- |
> > > > > | QAT | 2.7e-2 |  3.7e-2   | 2.8e-2  |  3.9e-2   |  4.5e-2   |  5.3e-2   |  6.8e-2   |  7.9e-2   |  8.3e-2   |  9.1e-2   |  6.9e-2   |  1.9e-2   |
> > > > > | REM | 3.6e-2 | 4.7e-2 | 5.6e-2 | 7.4e-2 | 9.9e-2 | 1.3e-1 | 1.5e-1 | 1.7e-1 | 2.0e-1 | 1.9e-1 | 1.4e-1 | 5.6e-2 |
> > > > > | MERM-S | 3.0e-2 | 4.2e-2 | 3.0e-2 | 4.5e-2 | 5.7e-2 | 7.0e-2 | 9.3e-2 | 1.0e-1 | 1.1e-1 | 1.2e-1 | 8.3e-2 | 2.5e-2 |
> > > > > | MREM-P | 2.9e-2 | 4.2e-2 | 3.3e-2 | 4.6e-2 | 5.9e-2 | 7.2e-2 | 9.4e-2 | 1.0e-1 | 1.2e-1 | 1.4e-1 | 8.8e-2 | 2.6e-2 |
> > > > >
> > > > > According to the table above, the gap of reconstruction error between QAT and MREM-S/P becomes larger in deeper transformer layers (except for last two layers as explained in L302). This matches the intuition as the reconstruction error accumulates and full end-to-end training with QAT can solve this issue better. Nonetheless, MREM-S/P are much closer to QAT at each transformer layer when compared with REM, indicating the advantages of our module-wise reconstruction error minimization in out approach.
> > > > >
> > > > > As there is little time left now we only present the table results here, but we will re-draw Fig 5c & 5d to include QAT in the revision. We hope the results above can address your concern. Many thanks again for your comments.

---

> ### Author Response · Authors · 2022-08-09
> **Kindly reminder for response**
>
> As the deadline for the rebuttal discussion is approaching, please let us know whether our replies have addressed your concerns. Please contact us if any further clarifications are required.

---

> > ### Comment · Area_Chair_o9XH · 2022-08-09
> > **Additional thoughts or revisions following author response?**
> >
> > Reviewer zLnQ, it would be great if you could follow up with any additional thoughts following the authors' response to your review. Thanks!

---

### Official Review · Reviewer_LBus · 2022-07-16

**Rating:** 7
**Confidence:** 3
**Soundness:** 3 good
**Presentation:** 3 good
**Contribution:** 3 good

**Summary:**

This paper presents a method of quantizing the parameters of pre-trained language models by dividing the model parameters into sets (or modules) and quantizing each of them separately via techniques similar to distillation referred to as "quantization error minimization". To improve the speed of this process, the authors propose learning parameters of each module in parallel (even though in general the output of one module is the input to the next) by (1) maintaining a cache of old activation values and asynchronously updating their values and using them as desired by each module, (2) using "teaching-forcing" by providing the original activation from the unquantized model in the beginning to reduce error propagation.

With experiments on quantizing BERT-base and BERT-large to different bit-sizes with downstream evaluation on GLUE and SQuAD, the authors show improvements over baselines in terms of training speed and memory required while using less data for quantization and mostly maintaining task performances.


**Questions:**

See weaknesses.

**Strengths And Weaknesses:**

Strengths:

1. The proposed solution is simple, where each component--module-based minimization, teaching forcing, and parallel computation--has sound motivation and leads to empirical improvements.
2. Clear improvements in training time and memory overhead compared to baselines.

Weaknesses:
1. Choice of #Bits for baselines and proposed methods (In table 3) are not the same, doesn't that make the comparison unfair?
2. The proposed method still lags behind QAT. Even though they require more data and more training time, this is a one-time investment and if a quantized model however obtained performs better downstream, why would it not be preferable? While data accessibility is a valid concern, other concerns listed in 2.2 are not convincing in favor of using PTQ.

---

> ### Author Response · Authors · 2022-08-02
> **To Reviewer LBus**
>
> Many thanks for your insightful comments and valuable questions!
>
> - **Q1**: "Choice of #Bits for baselines and proposed methods (In table 3) are not the same, doesn't that make the comparison unfair."
>
>   **A1**: Thanks for your question. We have included the baseline results from their original papers that have the most overlap with our setting. While our bit configuration (i.e., W2-E2-A4) is the lowest among these baselines, we can still obtain competitive results.
>
>   Furthermore,  we also newly add comparisons with REM (Q2 from Reviewer JA5e) and other PTQ methods (i.e. BRECQ and QDrop) (Q2 from Reviewer Z1iD), and our proposed MREM also achieves better performance with the same or lower #Bits.
>
> - **Q2**: "The proposed method still lags behind QAT. Even though they require more data and more training time, this is a one-time investment and if a quantized model however obtained performs better downstream, why would it not be preferable? While data accessibility is a valid concern, other concerns listed in 2.2 are not convincing in favor of using PTQ."
>
>   **A2**: This is a very interesting question with regard to the nature of PTQ. While data accessibility is one of the concerns, training time and memory also matter.
>
>   (1) **Training time**: Compared with QAT, PTQ can significantly reduce the production cycle [1],  which can be helpful to quantize models that are frequently updated in some scenarios (e.g., recommendation or information retrieval with streaming data). Meanwhile, the rapid training also enables researchers to explore more quantization configurations.
>
>   (2)  **Memory saving**: Our approach allows large pretrained language models to be post-training quantized on cheaper devices (e.g. BERT-large on a single NVIDIA GTX 1080 Ti as shown in Figure 2.b). While this is a not big concern when adequate computation is available, this allows a new alternative for researchers without many computation resources. In addition, when the accuracy is similar, saving the training time and memory also translates to lower financial costs and carbon emissions.
>
> [1] Li, et al. Brecq: Pushing the limit of post-training quantization by block reconstruction. ICLR, 2021.
>
> [2] Wei, et al. QDrop: Randomly Dropping Quantization for Extremely Low-bit Post-Training Quantization. ICLR 2022.

---

### Author Response · Authors · 2022-08-02
**The general response**

We sincerely thank all reviewers for their insightful comments and suggestions. We have carefully revised our paper (colored in blue), and here is the summary of the main revisions:

- In Appendix E, we provide more comparisons with REM, BRECQ and QDrop on GLUE tasks.

- In Appendix F, we provide more details on the quantization functions, as well as their implementation details.

In the response below, we use the line number based on the current revised version.

---

### Meta-Review · Area_Chair_o9XH · 2022-08-29

**Recommendation:** Accept
**Confidence:** Certain

**Metareview:**

In this paper the authors propose a practical method for post-training quantization (PTQ) of language models that divides the parameters and quantizes each separately (following BRECQ) in parallel, with asynchronous updates and a teacher forcing method to reduce error propagation. They show improvements on GLUE and SQuAD benchmarks. Reviewers agreed that the paper represents a solid practical contribution with convincing results, advancing the LM PTQ literature. The authors did a good job of addressing concerns and providing further analysis in the rebuttal.

**Award:**

No

---

### Decision · Program_Chairs · 2022-09-14

Accept